# Dual Progressive Prototype Network for Generalized Zero-Shot Learning

**Chaoqun Wang**[1 2]**, Shaobo Min**[3]**, Xuejin Chen**[1 2][*]**, Xiaoyan Sun**[2]**, Houqiang Li**[1 2]

[1]School of Data Science
[2]The National Engineering Laboratory for Brain-inspired Intelligence Technology and Application
University of Science and Technology of China, Hefei, Anhui, China
[3]Tencent Data Platform, Shenzhen, Guangdong, China
cq14@mail.ustc.edu.cn,bobmin@tencent.com,{xjchen99,sunxiaoyan,lihq}@ustc.edu.cn

## Abstract

Generalized Zero-Shot Learning (GZSL) aims to recognize new categories with auxiliary semantic information, *e.g.,* category attributes. In this paper, we handle the critical issue of domain shift problem, *i.e.*, confusion between seen and unseen categories, by progressively improving cross-domain transferability and category discriminability of visual representations. Our approach, named Dual Progressive Prototype Network (DPPN), constructs two types of prototypes that record prototypical visual patterns for attributes and categories, respectively. With attribute prototypes, DPPN alternately searches attribute-related local regions and updates corresponding attribute prototypes to progressively explore accurate attribute-region correspondence. This enables DPPN to produce visual representations with accurate attribute localization ability, which benefits the semantic-visual alignment and representation transferability. Besides, along with progressive attribute localization, DPPN further projects category prototypes into multiple spaces to progressively repel visual representations from different categories, which boosts category discriminability. Both attribute and category prototypes are collaboratively learned in a unified framework, which makes visual representations of DPPN transferable and distinctive. Experiments on four benchmarks prove that DPPN effectively alleviates the domain shift problem in GZSL.

## 1 Introduction

Deep learning methods depend heavily on enormous manually-labelled data, which limits their further applications [7, 8, 17, 52, 36, 19, 11]. Therefore, Generalized Zero-Shot Learning (GZSL) recently attracts increasing attention, which aims to recognize images from novel categories with only seen domain training data. Due to unavailable unseen domain data during training, GZSL methods introduce category descriptions, such as category attributes [15, 14] or word embedding [5, 37, 26, 41], to associate two domain categories.

A basic framework of embedding-based GZSL is to align global image representations with corresponding category descriptions in a joint embedding space [16, 3, 55, 38, 2, 6], as shown in Fig. 1 (a). Due to the domain shift problem across two domain categories, unseen domain images tend to be misclassified as seen categories. To address this issue, recent methods focus on discovering discriminative local regions to capture subtle differences between two domain categories. For example, AREN [45] and VSE [55] leverage attention mechanism to discover important part regions, which improves feature discrimination. DAZLE [20] and RGEN [46] introduce semantic guidance, *e.g.*, category

---

[*]Corresponding Author

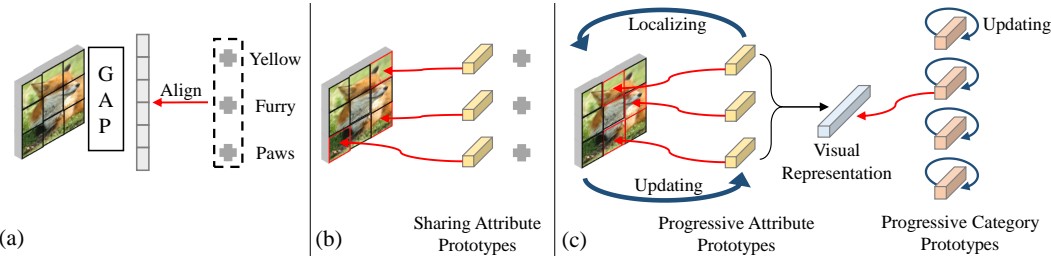

Figure 1: The motivation of DPPN. (a) General GZSL methods directly align global image features with category attributes. (b) A typical part-based method, *i.e.*, APN [47], learns prototypes shared by all images for attribute localization. (c) DPPN progressively adjusts prototypes according to different images and introduces category prototypes to enhance category discriminability.

attributes, into region localization to narrow the semantic-visual gap. Among existing methods, APN [47] is most related to our approach. As shown in Fig. 1 (b), APN constructs visual prototypes to indicate the typical visual patterns of each attribute, for example describing what attribute "Furry" visually refers to, and these prototypes are shared across all images to search attribute-matched local regions. However, due to image variances, the textures corresponding to the same attribute may vary seriously across images. Thus, sharing prototypes in APN can not well depict the target image.

In this paper, we propose a novel Dual Progressive Prototype Network (DPPN), which constructs two types of progressive prototypes for respective attributes and categories to gradually improve cross-domain transferability and category discriminability of visual representations. Instead of sharing prototypes, DPPN dynamically adjusts attribute prototypes for each image to capture the vital visual differences of the same attribute in different images. This is achieved by alternately localizing attribute regions and updating attributes prototypes in turn, as shown in Fig. 1 (c). With image-specific prototypes, attribute localization, *i.e.*, attribute-region correspondence, gets more accurate. To explicitly preserve such correspondence in the final representations, DPPN aggregates the attribute-related local features by concatenation, instead of widely-used Global Average Pooling (GAP) that will damage the attribute localization ability. Furthermore, along with progressively-updated attribute prototypes, DPPN also builds category prototypes to record prototypical visual patterns for different categories. The category prototypes are projected into multiple spaces to progressively enlarge category margins, strengthening category discriminability of visual representations. Consequently, with cross-domain transferability and category discriminability, DPPN can effectively bridge the gap between seen and unseen domains.

Experiments on four benchmarks demonstrate that our DPPN alleviates the domain shift problem in GZSL and obtains new state-of-the-art performance. Our contributions can be summarized as three-fold. a) We propose a novel Dual Progressive Prototype Network (DPPN) that constructs progressive prototypes for both attributes and categories to gradually improve cross-domain transferability and category discriminability of visual representations. b) An alternation updating strategy is designed to dynamically adjust attribute prototypes according to target images. Besides, DPPN aggregates attribute-related local features by concatenation to produce image representations, which explicitly preserves the attribute-region correspondence. c) DPPN projects category prototypes into multiple spaces to progressively enhance category discriminability.

## 2 Related Work

**Generalized Zero-Shot Learning.** GZSL aims to recognize new categories using semantic knowledge transferred from seen categories. Early GZSL methods learn a joint embedding space to align global image representations with corresponding category descriptions, *e.g.,* attributes [15, 14, 35] or text descriptions [5, 37]. Since unseen and seen categories share a common semantic space, the semantic-aligned image representations can be transferred from seen to unseen domain. Based on this paradigm, many works focus on improving the discrimination of embedding space by designing elaborate semantic-visual alignment functions [38, 44, 1, 54, 3, 55, 27]. For example, some methods [40, 53, 55, 21, 33] use high-dimensional visual features to span the embedding space, which is

proved more discriminative than that spanned by category attributes. Other methods [23, 9, 42, 32] utilize auto-encoders to preserve semantic relationships between categories in the embedding space.

Though effective, these methods suffer from the domain shift problem, *i.e.,* two domains have different data distributions. Since only seen domain images are available during training, images from unseen categories tend to be recognized as seen categories. To this end, DVBE [31] and Boundary-based OOD [10] explore out-of-distribution detection to treat seen and unseen domains separately. Some works [28, 20] suppress the seen category confidence when recognizing images to better distinguish two domain samples. These methods can effectively alleviate the domain shift problem via extra processing, but they ignore the discriminability of local attribute-related information in distinguishing two domains.

**Part-Based GZSL.** Since global image representations contain much noisy background information which is trivial for knowledge transfer, recent part-based methods [49, 45, 47] aim to localize part regions and capture important visual details to better understand the semantic-visual relationship. For example, S$^2$GA [49], AREN[45], and VSE [55] leverage the attention mechanism to learn semantic-relevant representations by automatically discovering discriminative parts in images. RGEN [46] uses the region graph to introduce region-based relation reasoning to GZSL and learns complementary relationships between different region parts inside an image. GEM-ZSL [30] imitates human attention and predicts human gaze location to learn visual attention regions. Usually, the semantic guidance, *e.g.,* category attributes, is used to guide the part localization [29]. Thus their generated local features can better match corresponding category attributes, which can alleviate the domain shift problem. Instead of treating category attributes as an all-in-one vector, DAZLE [20] proposes a dense attribute attention mechanism to produce local attention for each attribute separately. APN [47] constructs prototypes for separate attributes, which are shared by all images to localize attribute-related regions via region searching. In this paper, instead of using prototypes shared by all images, DPPN dynamically adjusts attribute prototypes according to different images, which learns visual representations with more accurate attribute localization and transferability. Besides, in DPPN, we also design progressive category prototypes to enhance category discriminability of visual representations.

# 3 Dual Progressive Prototype Network

## 3.1 Problem Formulation

The target of GZSL is to recognize images of novel categories trained with only seen domain data. In this paper, we denote $\mathcal{S} = \{X, y, \boldsymbol{a}_y | X \in \mathcal{X}_s, y \in \mathcal{Y}_s, \boldsymbol{a}_y \in \mathcal{A}_s\}$ as seen domain data, where $X \in \mathbb{R}^{C \times N}$ indicates image features extracted by the backbone network, and $X_n \in \mathbb{R}^{C \times 1}$ encodes the local information at the $n$-th region. $y$ is the corresponding category label, and $\boldsymbol{a}_y \in \mathbb{R}^{N_a \times 1}$ is the category description, such as the category-level vector with $N_a$ attributes. $C$ is the number of feature channels, and $N = W \times H$. The unseen domain data is similarly defined as $\mathcal{U}$, and $\mathcal{Y}_s \cap \mathcal{Y}_u = \phi$. Given $\mathcal{S}$ during training, GZSL aims to recognize images from either $\mathcal{X}_s$ or $\mathcal{X}_u$ during inference. A basic framework is to learn an image representation $f(X)$ that is aligned with corresponding category attributes by minimizing:

$$\mathcal{L}_{v2s} = - \sum_{X \in \mathcal{X}_s} \log \frac{\exp(f(X)^{\mathrm{T}} \boldsymbol{a}_y)}{\sum_{j \in \mathcal{Y}_s} \exp(f(X)^{\mathrm{T}} \boldsymbol{a}_j)}, \tag{1}$$

where $f(\cdot)$ is a visual projection function, which is generally implemented via Global Average Pooling (GAP) and linear projection. $v2s$ is the abbreviation of visual-to-semantic projection.

Based on $\mathcal{L}_{v2s}$, APN [47] expects to improve the localization ability of attributes for intermediate feature $X$. Thus, APN constructs a set of attribute prototypes $\mathcal{P} = \{\boldsymbol{p}_1, \cdots, \boldsymbol{p}_{N_a}\}$, where $\boldsymbol{p}_i \in \mathbb{R}^{C \times 1}$ records visual patterns for the $i$-th attribute, *e.g.,* depicting what attribute "Yellow Wing" looks like. $\mathcal{P}$ is learnable and shared by all images. With $\mathcal{P}$, APN learns attribute localization by minimizing:

$$\mathcal{L}_{apn} = \sum_{x \in \mathcal{X}_s} d(l(X, \mathcal{P}), \boldsymbol{a}_y). \tag{2}$$

$l(X, \mathcal{P})$ is an attribute localization function that searches the most related local feature $X_n$ for $\boldsymbol{p}_i \in \mathcal{P}$, which regresses attributes $\hat{\boldsymbol{a}}_y \in \mathbb{R}^{N_a \times 1}$. $d(\cdot, \cdot)$ is a distance measurement function, *e.g.,* $l_2$ norm, that

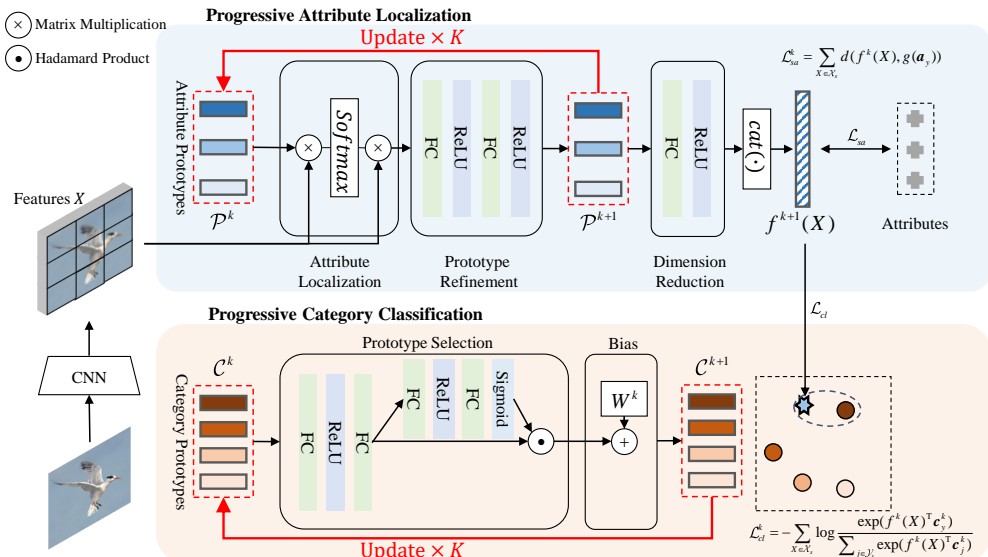

Figure 2: Training pipeline of DPPN. DPPN progressively learns attribute prototypes $\mathcal{P}^k$ and category prototypes $\mathcal{C}^k$ via $\mathcal{L}_{sa}$ and $\mathcal{L}_{cl}$. $\{f^k(X)|k = 1, \cdots, K\}$ consist of visual representations of $K$ iterations, which gradually capture attribute localization and category discrimination.

aligns the predicted attributes $\hat{\boldsymbol{a}}_y$ and ground truth attributes $\boldsymbol{a}_y$. Finally, the main insight of APN is to minimize $\mathcal{L}_{apn} + \mathcal{L}_{v2s}$, and the inference function is:

$$\hat{y} = \arg \min_{y \in \mathcal{Y}_s \cup \mathcal{Y}_u} d(f(X), \boldsymbol{a}_y). \tag{3}$$

However, due to instance variance, occlusion, and noise, the visual textures that correspond to the same attribute may vary severely across different images. Thus, it is unreasonable to only rely on sharing attribute prototypes $\mathcal{P}$ to accurately localize attribute-related regions for each individual image. Besides, $\mathcal{L}_{apn}$ is applied to intermediate features $X$ as a mere constraint. During inference, the final visual representation $f(X)$ for recognition is directly aggregated by the intermediate features using GAP, which damages the attribute localization ability.

To this end, our Dual Progressive Prototype Network (DPPN) progressively explores attribute localization and category discrimination for different images with two modules, *i.e.,* Progressive Attribute Localization and Progressive Category Classification, as shown in Fig. 2.

### 3.2 Progressive Attribute Localization

The Progressive Attribute Localization (PAL) module aims to dynamically adjust attribute prototypes according to the target image to progressively capture local correspondence between different attributes and image regions.

Define $\mathcal{P}^0 = \{\boldsymbol{p}_1^0, \cdots, \boldsymbol{p}_{N_a}^0\}$ as a set of trainable attribute prototypes, which are randomly initialized and shared across all images. With $\mathcal{P}^0$ initialized, PAL first localizes related local regions for each prototype $\boldsymbol{p}_i^0$, and then calculates specific visual features for all $\{\boldsymbol{p}_1^0, \cdots, \boldsymbol{p}_{N_a}^0\}$ by:

$$\mathcal{P}^1 = f_{ar}(XS), S = \hbar(X^{\mathrm{T}}\mathcal{P}^0), \tag{4}$$

where $S \in \mathbb{R}^{N \times N_a}$ is a similarity matrix, and $S_{n,i}$ measures the similarity between the $n$-th local feature $X_n$ and the $i$-th attribute prototype $\boldsymbol{p}_i^0$. $\hbar(\cdot)$ is a Softmax normalization along each column. With $S$, Eq. (4) aggregates related region features in $X$ to calculate attribute-specific features, which produces $\mathcal{P}^1 = \{\boldsymbol{p}_1^1, \cdots, \boldsymbol{p}_{N_a}^1\}$. $f_{ar}(\cdot)$ is a prototype refinement function implemented by two fully-connected (FC) layers as shown in Fig. 2. Compared to the original image feature $X \in \mathbb{R}^{C \times N}$, $\mathcal{P}^1 \in \mathbb{R}^{C \times N_a}$ explicitly captures specific visual patterns of the target image for each attribute, *e.g.,* $\boldsymbol{p}_i^1$ aggregates related local features in $X$ that correspond to the $i$-th attribute.

Considering instance variance, the sharing attribute prototypes $\mathcal{P}^0$ cannot well localize accurate attribute-related regions for all images. In Eq. (4), compared to $\mathcal{P}^0$, the visual patterns of $\mathcal{P}^1$, that correspond to different attributes, are more specific to the given image feature $X$. Thus, PAL further regards $\mathcal{P}^1$ as updated attribute prototypes from $\mathcal{P}^0$ for the target image. By replacing $\mathcal{P}^k$ with $\mathcal{P}^{k+1}$ and repeating Eq. (4), PAL can progressively adjust attribute prototypes for a specific image by:

$$\mathcal{P}^{k+1} = f_{ar}\big(X\hbar(X^{\mathrm{T}}\mathcal{P}^k)\big), \tag{5}$$

where $\mathcal{P}^{k+1}$ leads to better attribute localization than $\mathcal{P}^k$.

For the $k$-th iteration, since $\mathcal{P}^k \in \mathbb{R}^{C \times N_a}$ contains specific visual patterns for different attributes, PAL concatenates all $\{\boldsymbol{p}_1^k, \cdots, \boldsymbol{p}_{N_a}^k\}$ to produce the visual representation $f^k(X)$:

$$f^k(X) = cat\big(f_{rd}(\mathcal{P}^k)\big), \tag{6}$$

where $f_{rd}(\cdot)$ is a dimension reduction layer and projects each $\boldsymbol{p}_i^k \in \mathbb{R}^{C \times 1}$ into $\mathbb{R}^{D \times 1}$, where $D < C$, to avoid excess calculation complexity. $cat(\cdot)$ concatenates all elements of the input. $f^k(X) \in \mathbb{R}^{N_v \times 1}$, where $N_v = D \times N_a$. With multi-iterative $\{\mathcal{P}^k | k = 1, \cdots, K\}$, PAL gradually generates $K$ visual representations $\{f^k(X) | k = 1, \cdots, K\}$.

Finally, $f^k(X)$ is aligned with corresponding attributes in a joint embedding space by:

$$\mathcal{L}_{sa}^k = \sum_{X \in \mathcal{X}_s} d\big(f^k(X), g(\boldsymbol{a}_y)\big), \tag{7}$$

where $g(\cdot)$ is a semantic projection function implemented by FC to project attribute vector into a latent space, where the visual representations and projected attribute features can be well aligned, following [31]. The semantic alignment supervision $\mathcal{L}_{sa}$ is applied to all $\{f^k(X) | k = 1, \cdots, K\}$ for training acceleration. As $k$ increases appropriately, $f^k(X)$ localizes attributes more accurately. Thus, $f^K(X)$ is used as the final visual representation for inference.

Consequently, by dynamically adjusting attribute prototypes according to the target image, PAL can progressively improve the attribute localization ability of visual representations, as shown in Fig. 3. PAL captures the attribute-region correspondence, which narrows the semantic-visual gap between category attributes and visual representations and boosts knowledge transfer between seen and unseen domains.

### 3.3  Progressive Category Classification

Besides exploring the correspondence between attributes and local image regions via PAL, we design a Progressive Category Classification (PCC) module to repel visual representations from different categories, which can enlarge category margins.

Similar to the sharing attribute prototypes in PAL, PCC defines a set of learnable category prototypes $\mathcal{C}^0 = \{\boldsymbol{c}_1^0, \cdots, \boldsymbol{c}_{N_c}^0\}$, where $\boldsymbol{c}_j^0 \in \mathbb{R}^{N_v \times 1}$ records the visual representation center for the $j$-th category. $N_c = |\mathcal{Y}_s|$ is the number of seen categories. Since the attribute prototypes are progressively updated in PAL and $K$ visual representations $\{f^k(X) | k = 1, \cdots, K\}$ are accordingly generated for each image, sharing category prototypes $\mathcal{C}^0$ cannot well model visual category differences for all iterations of $f^k(X)$. Thus, PCC is similarly designed to adjust category prototypes for different $f^k(X)$ by:

$$\mathcal{C}^{k+1} = f_{cs}(\mathcal{C}^k) + W^k, \tag{8}$$

where $f_{cs}(\cdot)$ is a prototype selection function as shown in Fig. 2. Since $f^{k+1}(X)$ derives from $f^k(X)$, the category center at the $(k+1)$-th iteration should not deviate from that of the $k$-th iteration. Thus, $f_{cs}(\cdot)$ actually serves as a gating function implemented by channel attention mechanism, which controls the information flow from $\mathcal{C}^k$ to $\mathcal{C}^{k+1}$. This can ease the training difficulty of PCC by avoiding repetitive learning for $\mathcal{C}^{k+1}$. $W^k$ is a learnable bias at the $k$-th iteration, which supplements some specific information for $\mathcal{C}^{k+1}$.

At the $k$-th iteration, with the visual representation $f^k(X)$ and category prototypes $\mathcal{C}^k = \{\boldsymbol{c}_1^k, \cdots, \boldsymbol{c}_{N_c}^k\}$, PCC repels different categories by:

$$\mathcal{L}_{cl}^k = -\sum_{X \in \mathcal{X}_s} \log \frac{\exp(f^k(X)^{\mathrm{T}} \boldsymbol{c}_y^k)}{\sum_{j \in \mathcal{Y}_s} \exp(f^k(X)^{\mathrm{T}} \boldsymbol{c}_j^k)}. \tag{9}$$

Similar to $\mathcal{L}_{sa}^k$, $\mathcal{L}_{cl}^k$ is applied to all the $K$ iterations. Compared to $\mathcal{L}_{v2s}$ in Eq. (1), $\mathcal{L}_{cl}$ can better repel visual representations from different categories via progressive category prototypes $\{\mathcal{C}^k | k = 1, \cdots, K\}$.

With progressively-updated category prototypes, PCC improves the category discriminability of visual representations $\{f^k(X) | k = 1, \cdots, K\}$.

### 3.4 Overall Objective

Overall, the objective loss function of DPPN is:

$$\mathcal{L}_{all} \leftarrow \sum_{k=1}^{K} (\mathcal{L}_{sa}^k + \lambda \mathcal{L}_{cl}^k), \tag{10}$$

where $\lambda$ is the hyper-parameter to balance $\mathcal{L}_{cl}$. The attribute prototypes and category prototypes are collaboratively trained in a unified framework, which enables the final visual representation $f^K(X)$ to simultaneously capture attribute-region correspondence and category discrimination. During inference, only visual representation $f^K(X)$ at the $K$-th iteration is used by:

$$\hat{y} = \arg \min_{y \in \mathcal{Y}_s \cup \mathcal{Y}_u} d(f^K(X), g(\boldsymbol{a}_y)). \tag{11}$$

### 3.5 Discussion

Compared to APN [47], DPPN is a much different and novel method with three main differences: a) instead of sharing prototypes for all images in APN, DPPN dynamically adjusts attribute prototypes according to different images. Specifically, DPPN introduces attribute-related clues from the target image feature into attribute prototypes, so that the prototypes are more adapted to the target image and result in better attribute localization; b) different from APN's averagely pooling local visual features into a global one, DPPN concatenates local features to represent an image, which better preserves attribute-region correspondence. The final representation of DPPN is made up of attribute-localized features during both training and inference, instead of regarding attribute localization as mere supervision during training in APN; and c) DPPN further exploits progressive category prototypes to repel visual representations from different categories, which enhances category discrimination.

## 4 Experiments

### 4.1 Experimental Settings

**Datasets.** Four public GZSL benchmarks, *i.e.*, Caltech-USCD Birds-200-2011 (CUB) [43], SUN [35], Animals with Attributes2 (AWA2) [24], and Attribute Pascal and Yahoo (aPY) [14], are adopted in this paper. CUB contains $11,788$ bird images in 200 species with 312 description attributes. SUN contains $14,340$ scene images in 717 classes with 102 attributes. AWA2 contains $37,322$ animal images in 50 classes with 85 attributes. aPY contains $15,339$ object images in 32 categories with 64 attributes.

**Evaluation Metrics.** The widely-used harmonic mean $H = (2MCA_u \times MCA_s)/(MCA_u + MCA_s)$ is used to evaluate GZSL performance. $MCA_s$ and $MCA_u$ are the Mean Class Top-1 Accuracy for seen and unseen domains, respectively.

**Implementation Details.** The input images are resized to $448 \times 448$ following [55, 45]. Random cropping and flipping are used for data augmentation. ResNet-101 [18] pretrained on ImageNet [12] is used as the backbone. A two-step training strategy is adopted, which trains DPPN with the fixed backbone and then fine-tunes the whole network on two 1080ti GPUs. Adam optimizer [22] is used with batch size of 64 and $lr = 2e - 4$. $C = 512$ since we use a conv. layer for dimension reduction after the backbone. $N_a$ is the number of attributes, which is 312, 85, 64, 102 for respective CUB, AWA2, aPY, and SUN. $K = 3$ and $\lambda = 1.0$. $N_v$ will be discussed in the ablation study.

### 4.2 Ablation Study

**Analysis of Attribute Localization and Category Discrimination.** The core motivation of DPPN is to learn visual representations that simultaneously explore category discrimination and attribute-

| Method | $\mathcal{L}_{v2s}$ | $\mathcal{L}_{cl}$ | $\mathcal{L}_{sa}$ | $K$ | CUB | | | aPY | | | GFLOPs |
|---|---|---|---|---|---|---|---|---|---|---|---|
| | | | | | $MCA_u$ | $MCA_s$ | $H$ | $MCA_u$ | $MCA_s$ | $H$ | |
| Base-V2S | ✓ | | | - | 50.5 | 84.4 | 63.2 | 30.4 | 42.6 | 35.5 | 62.396 |
| +PCC | | ✓ | | 1 | 59.3 | 70.2 | 64.3 | 33.2 | 50.9 | 40.2 | 62.440 |
| +PAL | | | ✓ | 1 | 64.7 | 71.8 | 68.1 | 34.2 | 53.8 | 41.8 | 62.809 |
| +PCC&PAL | | ✓ | ✓ | 1 | 69.2 | 71.4 | 70.3 | 35.6 | 59.0 | 44.4 | 62.842 |
| +PCC&PAL | | ✓ | ✓ | 3 | 70.2 | 77.1 | 73.5 | 40.0 | 61.2 | 48.4 | 62.900 |

Table 1: Effect of PCC and PAL on CUB and aPY datasets. GFLOPs is calculated with input size $448 \times 448$ on the CUB dataset.

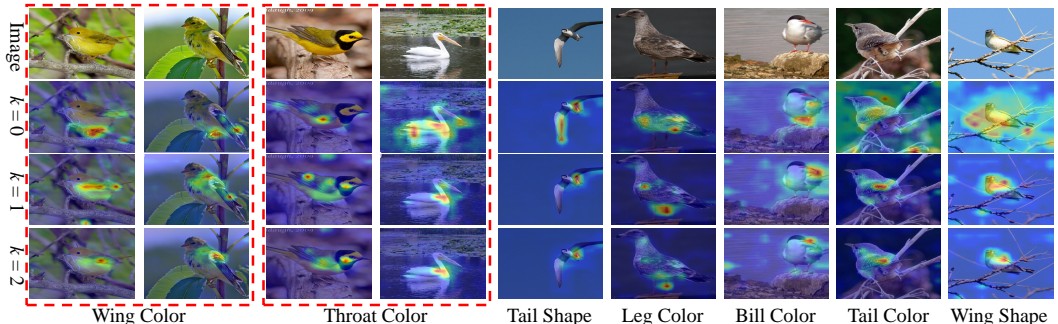

Figure 3: Visualization of attribute localization at different iterations. The localization gets more and more accurate as $k$ increases from 0 to 2.

region correspondence via the proposed PCC and PAL, thus we analyse how PCC and PAL affect GZSL performance in this part. For simple comparison, we set $K = 1$ in PCC and PAL. Results are listed in Table 1. "Base-V2S" is the baseline method trained by $\mathcal{L}_{v2s}$ in Eq. (1). "+PCC" adds PCC module to Base-V2S, which introduces stronger category discrimination constraint $\mathcal{L}_{cl}$. We can observe that "+PCC" obtains $1.1\%$ and $4.7\%$ gains on $H$ over "Base-V2S" on CUB and aPY. This is because that PCC explicitly pushes representations away from different categories via category prototypes, thus enlarging category margins for more accurate category classification and boosting GZSL. The third model "+PAL" replaces $f(\cdot)$ in "Base-V2S" with PAL, which enables visual representations with attribute localization ability. Compared with "Base-V2S", PAL module brings $4.9\%$ and $6.3\%$ gains on CUB and aPY. This derives from that PAL captures attribute-region correspondence by utilizing attribute prototypes to localize attribute-related local regions and produce attribute-specific visual representations. Finally, "+PCC&PAL" simultaneously considers category discrimination and attribute localization by incorporating both PCC and PAL, which obtains $7.1\%$ and $8.9\%$ gains. This proves that both attribute-region correspondence and category discrimination are critical to GZSL and complementary to each other.

Notably, PAL and PCC bring negligible additional computation, even when $K = 3$, because $f_{rd}(\cdot)$ in Eq. (6) controls the dimension of representation $f^k(X)$ to limit the computation burden. We visualize the difference of representation distribution between Base-V2S and our DPPN in the Appendix.

**Effect of Progressive Prototype Updating.** DPPN progressively updates attribute and category prototypes to learn more transferable and distinctive representations. Here, we analyse how such a progressive learning strategy impacts the attribute localization ability and category recognition by evaluating varying $K$ in PAL and PCC. The results are given in Fig. 4.

As $K$ rises from 1 to 3, $H$ on all the four datasets gradually increases. The best $H = 73.5\%$, $73.1\%$, $48.4\%$, and $41.0\%$ on CUB, AWA2, aPY, and SUN is obtained when $K = 3$. This demonstrates that, with category and attribute prototypes updated, the visual representations become more discriminative and transferable to the unseen categories. Here, to intuitively present the attribute localization progressively learned by PAL, we visualize the attribute localization results of PAL at different iteration $k$ when setting $K = 3$. As shown in Fig. 3, the localization gets more and more precise as attribute prototypes gradually update. With progressively updated attribute prototypes, the PAL module can finally accurately localize corresponding attribute-related visual regions. Besides, with updating, the prototype for the same attribute gets more specific to the target image in the first four columns, reflecting that progressive updating can adapt prototypes according to different images.

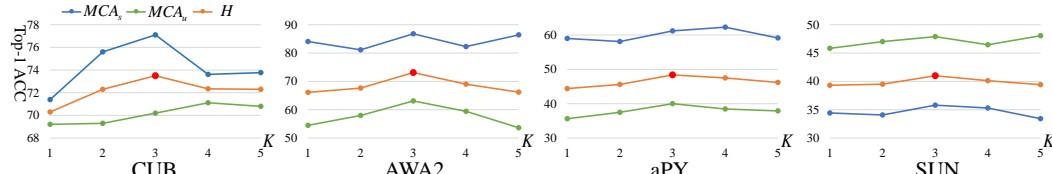

Figure 4: Effect of progressive updating with varying $K$ on four datasets.

| | PAL | | | | PCC | CUB | | | aPY | | |
|---|---|---|---|---|---|---|---|---|---|---|---|
| | $f_{ar}(\cdot)$ | $cat(\cdot)$ | $sum(\cdot)$ | $max(\cdot)$ | $f_{cs}(\cdot)$ | $MCA_u$ | $MCA_s$ | $H$ | $MCA_u$ | $MCA_s$ | $H$ |
| DPPN | ✓ | ✓ | | | ✓ | 70.2 | 77.1 | 73.5 | 40.0 | 61.2 | 48.4 |
| | | ✓ | | | ✓ | 67.4 | 78.2 | 72.4 | 36.3 | 60.9 | 45.5 |
| | ✓ | | ✓ | | ✓ | 67.0 | 70.5 | 68.7 | 35.6 | 62.0 | 45.2 |
| | ✓ | | | ✓ | ✓ | 70.4 | 73.0 | 71.7 | 37.3 | 59.4 | 45.8 |
| | ✓ | ✓ | | | | 68.3 | 76.0 | 71.9 | 38.4 | 57.1 | 45.9 |

Table 2: Evaluation of components in DPPN.

This proves that attribute prototypes can capture the attribute-region correspondence, and progressive updating makes prototypes more specific and distinctive. When $K > 3$, $H$ drops. The reason may be that the over-updated prototypes become unstable and hard to train. $K = 3$ is a good trade-off between general knowledge of a whole dataset and specific knowledge towards an image. Thus, we set $K = 3$ for the rest experiments.

In summary, both quantitative and qualitative results demonstrate that progressive updating can improve attribute and category prototypes, which better captures attribute-region correspondence and category discrimination.

**Evaluation of Components in PAL.** PAL aims to learn visual representations with accurate attribute localization. Thus, we evaluate two important components of PAL, *i.e.*, $f_{ar}(\cdot)$ in Eq. (5) and $cat(\cdot)$ in Eq. (6). The results are given in Table 2. $f_{ar}(\cdot)$ is a refinement function between $\mathcal{P}^k$ and $\mathcal{P}^{k+1}$ to improve prototype quality. As shown in the second row of Table 2, without $f_{ar}(\cdot)$, $H$ drops by 1.1% and 2.9% on CUB and aPY, respectively. This demonstrates that $f_{ar}(\cdot)$ benefits attribute prototype updating, thereby boosting attribute localization ability.

$cat(\cdot)$ is the aggregation function used to produce visual representation $f^k(X)$ by concatenating attribute prototypes $\mathcal{P}^k = \{\boldsymbol{p}_1^k, \cdots, \boldsymbol{p}_{N_a}^k\}$. Compared to summing $\{\boldsymbol{p}_1^k, \cdots, \boldsymbol{p}_{N_a}^k\}$ up or max pooling operation, $cat(\cdot)$ better preserves attribute-region correspondence. The third and fourth rows of Table 2 show that $H$ drops as replacing $cat(\cdot)$ with either summing up or max pooling. This proves that $cat(\cdot)$ benefits local correspondence preservation.

**Effect of $f_{cs}(\cdot)$ in PCC.** $f_{cs}(\cdot)$ in Eq. (8) serves as a gating function to ease the training of category prototypes. In this part, we analyse the impact of $f_{cs}(\cdot)$. In Eq. (8), $f_{cs}(\cdot)$ removes redundancy in $\mathcal{C}^k$. Without $f_{cs}(\cdot)$, the category prototype updating becomes $\mathcal{C}^{k+1} = \mathcal{C}^k + W^{k+1}$, which passes all information in $\mathcal{C}^k$ to $\mathcal{C}^{k+1}$. The results are listed in the last row of Table 2. Without $f_{cs}(\cdot)$, $H$ drops by 1.6% and 2.5% on CUB and aPY. This proves that $f_{cs}(\cdot)$ helps to ease the training process of PCC, which learns more discriminative visual representations.

**Effect of $\lambda$.** $\lambda$ is the hyper-parameter to balance $\mathcal{L}_{cl}$. Here, we evaluate the effect of $\lambda$ as shown in Fig. 5 (a) and (b). As $\lambda$ rises from 0.0 to 1.0, *i.e.*, category discrimination supervision $\mathcal{L}_{cl}$ is introduced into DPPN, $H$ increases on both CUB and aPY. The best $H$ is obtained when $\lambda = 1.0$. This proves the effectiveness of category discrimination brought by PCC. When $\lambda > 1.0$, $H$ starts to drop. Thus, we set $\lambda = 1.0$ for better results.

**Effect of $N_v$.** $N_v$ is the dimension of $f^k(X)$ and $N_v = D \times N_a$. $N_a$ is the number of attributes, which are 312, 85, 64, 102 for CUB, AWA2, aPY, and SUN, respectively. Here, we set different $D$ to evaluate the effect of $N_v$ on recognition performance and calculation addition. Fig. 5 (c) shows the values of $H$ as $N_v$ varies. Fig. 5 (d) shows GLOPs addition over $N_v = 512$. When $N_v$ is around 2,048, best performance is obtained with a relatively small calculation complexity addition. Thus, we set $N_v = 2496, 2125, 2048, 2142$ and $D = 8, 25, 32, 21$ for CUB, AWA2, aPY, and SUN, respectively.

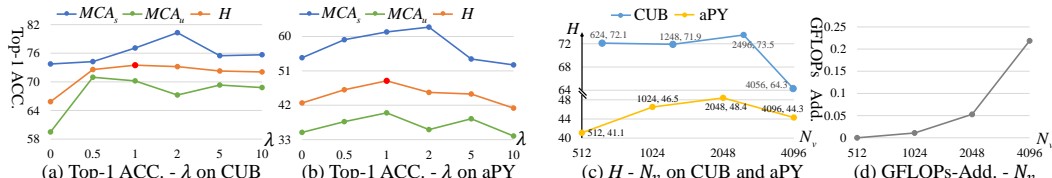

Figure 5: Effect of hyper-parameters $\lambda$ and $N_v$.

| | Methods | CUB | | | AWA2 | | | aPY | | | SUN | | |
|---|---|---|---|---|---|---|---|---|---|---|---|---|---|
| | | $MCA_u$ | $MCA_s$ | $H$ | $MCA_u$ | $MCA_s$ | $H$ | $MCA_u$ | $MCA_s$ | $H$ | $MCA_u$ | $MCA_s$ | $H$ |
| GEN. | IZF-Softmax[39] | 52.7 | 68.0 | 59.4 | 60.6 | 77.5 | 68.0 | 42.3 | 60.5 | 49.8 | 52.7 | 57.0 | 54.8 |
| | TF-VAEGAN[34] | 63.8 | 79.3 | 70.7 | - | - | - | - | - | - | 41.8 | 51.9 | 46.3 |
| | E-PGN[50] | 52.0 | 61.1 | 56.2 | 52.6 | 83.5 | 64.6 | - | - | - | - | - | - |
| | GCM-CF[51] | 61.0 | 59.7 | 60.3 | 60.4 | 75.1 | 67.0 | 37.1 | 56.8 | 44.9 | 47.9 | 37.8 | 42.2 |
| | CE-GZSL[17] | 63.9 | 66.8 | 65.3 | 63.1 | 78.6 | 70.0 | - | - | - | 48.8 | 38.6 | 43.1 |
| EMB. | MLSE[13] | 22.3 | 71.6 | 34.0 | 23.8 | 83.2 | 37.0 | 12.7 | 74.3 | 21.7 | 20.7 | 36.4 | 26.4 |
| | COSMO[4] | 44.4 | 57.8 | 50.2 | - | - | - | - | - | - | 44.9 | 37.7 | **41.0** |
| | PREN[48] | 32.5 | 55.8 | 43.1 | 32.4 | 88.6 | 47.4 | - | - | - | 35.4 | 27.2 | 30.8 |
| | VSE-S[55] | 33.4 | 87.5 | 48.4 | 41.6 | 91.3 | 57.2 | 24.5 | 72.0 | 36.6 | - | - | - |
| | LFGAA[29] | 43.4 | 79.6 | 56.2 | 50.0 | 90.3 | 64.4 | - | - | - | 20.8 | 34.9 | 26.1 |
| | AREN[45] | 63.2 | 69.0 | 66.0 | 54.7 | 79.1 | 64.7 | 30.0 | 47.9 | 36.9 | 40.3 | 32.3 | 35.9 |
| | CosineSoftmax[25] | 47.4 | 47.6 | 47.5 | 56.4 | 81.4 | 66.7 | 26.5 | 74.0 | 39.0 | 36.3 | 42.8 | 39.3 |
| | RGEN[46] | 73.5 | 60.0 | 66.1 | 76.5 | 67.1 | 71.5 | 48.1 | 30.4 | 37.2 | 31.7 | 44.0 | 36.8 |
| | DAZLE[20] | 56.7 | 59.6 | 58.1 | 60.3 | 75.7 | 67.1 | - | - | - | 52.3 | 24.3 | 33.2 |
| | APN[47] | 65.3 | 69.3 | 67.2 | 56.5 | 78.0 | 65.5 | - | - | - | 41.9 | 34.0 | 37.6 |
| | GEM-ZSL[30] | 64.8 | 77.1 | 70.4 | 64.8 | 77.5 | 70.6 | - | - | - | 38.1 | 35.7 | 36.9 |
| | **DPPN** | 70.2 | 77.1 | **73.5** | 63.1 | 86.8 | **73.1** | 40.0 | 61.2 | **48.4** | 47.9 | 35.8 | **41.0** |

Table 3: Results of GZSL on four classification benchmarks. Our DPPN belongs to embedding-based methods (EMB.). Generative methods (GEN.) utilize extra synthetic unseen domain data for training. The best result is bolded, and the second best is underlined.

## 4.3 Comparison with State-of-the-Art Methods

We compare DPPN with the state-of-the-art GZSL methods, of which the results are given in Table 3.

Among the existing methods, APN [47] is the most related method which also utilizes visual prototypes to localize visual parts. Different from APN that counts on only sharing prototypes for all images, our DPPN adjusts attribute prototypes dynamically according to the target image and exploits category prototypes to enhance category discrimination. Thus, DPPN learns specific and distinctive visual representations and surpasses APN by a large margin, *i.e.*, 6.3%, 7.6%, and 3.4% for $H$ on CUB, AWA2, and SUN datasets, respectively. APN does not provide codes and is approximately similar to PAL with $K = 1$. As shown in Fig. 3, it qualitatively implies that progressive updating attribute prototypes can learn visual representations with better attribute localization ability. Besides, DAZLE [20] uses dense attribute attention to focus on relevant regions, which is inferior to DPPN by 15.4%, 6.0%, and 7.8% on CUB, AWA2, and SUN, respectively. This proves the effectiveness of designing progressive attribute and category prototypes in GZSL.

Compared to other embedding-based methods, our method surpasses the best one by respectively 3.1%, 1.6%, and 9.4% for $H$ on CUB, AWA2, aPY datasets, and obtains comparable best $H$ performance on SUN dataset. The results demonstrate that progressively exploring attribute-region correspondence and category discrimination can effectively enhance cross-domain transferability and category discriminability of visual representations. The reason for relatively small improvement on SUN may be that #categories is large while #images in each category is small in SUN, leading to difficulties for DPPN to learn accurate attribute localization and category discrimination.

Compared to generative methods that utilize additional unseen category labels during training, DPPN can achieve comparable, even better results, especially on CUB and AWA2 datasets. This reveals that with the assistance of progressive attribute localization and category discrimination, DPPN can surpass generative methods without complex GAN training.

# 5 Conclusion

In this paper, we propose a Dual Progressive Prototype Network (DPPN) to progressively explore both attribute-region correspondence and category discrimination for GZSL. Specifically, DPPN constructs progressive prototypes for both attributes and categories. DPPN alternatively localizes attribute-related visual regions and adjusts attribute prototypes towards target images, which improves attribute localization ability and cross-domain transferability of visual representations. Along with progressive attribute prototypes, DPPN progressively projects category prototypes to multiple spaces to enforce visual representations away from different categories, thus enhancing category discriminability. Extensive experimental results on four public datasets demonstrate the effectiveness of our DPPN.

## Acknowledgments and Disclosure of Funding

This work was supported by National Natural Science Foundation of China (NSFC) under Grants 61632006 and 62076230.

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
