# Supplementary for Dual Progressive Prototype Network for Generalized Zero-Shot Learning

**Chaoqun Wang**[1 2], **Shaobo Min**[3], **Xuejin Chen**[1 2]*, **Xiaoyan Sun**[2], **Houqiang Li**[1 2]

[1]School of Data Science
[2]The National Engineering Laboratory for Brain-inspired Intelligence Technology and Application
University of Science and Technology of China, Hefei, Anhui, China
[3]Tencent Data Platform, Shenzhen, Guangdong, China
cq14@mail.ustc.edu.cn,bobmin@tencent.com,{xjchen99,sunxiaoyan,lihq}@ustc.edu.cn

## 1   Comparison without Post-Processing

Since some brand-new methods utilize post-processing, such as calibration stacking [5] or domain detector [2, 12], to alleviate the domain shift problem, we report the results of our Dual Progressive Prototype Network (DPPN) with post-processing in Table 3 of the main paper for fair comparison. In this part, we further compare our DPPN with recent methods that clearly report their results without post-processing, of which the comparison results are shown in Table 1. APN [15] only reports their results with calibration stacking. Our DPPN outperforms the best one by respectively 15.3%, 8.8%, and 7.3% for $H$ on CUB, AWA2, and aPY datasets, and obtains comparable performance on SUN dataset. This demonstrates the effectiveness of learning representations that progressively explore category discrimination and attribute-region correspondence.

| Methods | CUB | | | AWA2 | | | aPY | | | SUN | | |
|---|---|---|---|---|---|---|---|---|---|---|---|---|
| | $MCA_u$ | $MCA_s$ | $H$ | $MCA_u$ | $MCA_s$ | $H$ | $MCA_u$ | $MCA_s$ | $H$ | $MCA_u$ | $MCA_s$ | $H$ |
| SJE[1] | 23.5 | 59.2 | 33.6 | 8.0 | 73.9 | 14.4 | 3.7 | 55.7 | 6.9 | 14.7 | 30.5 | 19.8 |
| DEVISE[7] | 23.8 | 53.0 | 32.8 | 17.1 | 74.7 | 27.8 | 4.9 | 76.9 | 9.2 | 16.9 | 27.4 | 20.9 |
| SYNC[3] | 11.5 | 70.9 | 19.8 | 10.0 | 90.5 | 18.0 | 7.4 | 66.3 | 13.3 | 7.9 | 43.3 | 13.4 |
| MLSE[6] | 22.3 | 71.6 | 34.0 | 23.8 | 83.2 | 37.0 | 12.7 | 74.3 | 21.7 | 20.7 | 36.4 | **26.4** |
| LFGAA[10] | 36.2 | 80.9 | 50.0 | 27.0 | 93.4 | 41.9 | - | - | - | 18.5 | 40.0 | 25.3 |
| AREN[13] | 38.9 | 78.7 | 52.1 | 15.6 | 92.9 | 26.7 | 9.2 | 76.9 | 16.4 | 19.0 | 38.8 | 25.5 |
| DAZLE[8] | 42.0 | 65.3 | 51.1 | 25.7 | 82.5 | 39.2 | - | - | - | 21.7 | 31.9 | 25.8 |
| **DPPN** | 56.8 | 82.8 | **67.4** | 34.7 | 94.1 | **50.7** | 17.7 | 80.8 | **29.0** | 18.1 | 42.1 | 25.3 |

Table 1: Results of GZSL on four classification benchmarks under no post-processing settings. The best result is bolded.

## 2   Conventional ZSL Results

We provide the comparison results with several brand-new methods that are most related to DPPN under CZSL setting in Table 2. $MCA_u$ is used as evaluation metric. DPPN obtains comparable, even better performance, compared to SOTA methods, which demonstrates the transferability of our method. Notably, the performance of DPPN on CZSL is not as impressive as in GZSL. The reason is that DPPN focuses on improving representation discrimination to alleviate the domain bias problem.

---

*Corresponding Author

35th Conference on Neural Information Processing Systems (NeurIPS 2021).

| Methods | CUB | AWA2 | aPY | SUN |
|---|---|---|---|---|
| AREN[13] | 71.8 | 67.9 | 39.2 | 60.6 |
| CosineSoftmax[9] | 54.4 | 71.1 | 38.0 | 62.6 |
| RGEN[14] | 76.1 | **73.6** | 44.4 | **63.8** |
| DAZLE[8] | 65.9 | - | - | - |
| GEM-ZSL[11] | **77.8** | 67.3 | - | 62.8 |
| APN[15] | 72.0 | 68.4 | - | 61.6 |
| **DPPN** | **77.8** | 73.3 | **45.1** | 61.5 |

Table 2: Results of CZSL. $MCA_u$ is used as the evaluation metric. The best result is bolded, and the second best is underlined.

## 3 Comparison on Evaluation Metric AUSUC

In this section, we give the comparison results with recent related methods on the evaluation metric of AUSUC [4]. As shown in Table 3, our DPPN outperforms the best one by respectively $19.7\%$ and $7.7\%$ for AUSUC on CUB and AWA2 datasets, and obtains comparable performance on SUN dataset. The robust improvements over various metrics prove that DPPN can effectively alleviate the domain bias problem in GZSL.

| Methods | CUB | AWA2 | aPY | SUN |
|---|---|---|---|---|
| SYNC[3] | 33.7 | 50.4 | - | 24.1 |
| COSMO[2] | 35.7 | - | - | 23.9 |
| EXEM[4] | 36.6 | 55.9 | - | **25.1** |
| **DPPN** | **56.3** | **63.6** | **33.4** | 23.1 |

Table 3: Comparison results on AUSUC metric. The best result is bolded.

## 4 Results of Input Size $224 \times 224$

For fair comparison, we use the setting of input size $448 \times 448$ following recent SOTA methods, *e.g.*, VSE-S [18], GEM-ZSL [11], and AREN [13]. In this part, we conduct experiments comparing DPPN with recent methods under the setting of input size $224 \times 224$, of which the results are shown in Table 4. From the results, our DPPN outperforms the best previous method by respectively $3.8\%$, $6.7\%$, and $2.9\%$ on CUB, aPY, and SUN datasets, and achieves the second-best performance on AWA2. The SOTA performance on different resolutions demonstrates the effectiveness and generalization of our DPPN.

| Methods | CUB | | | AWA2 | | | aPY | | | SUN | | |
|---|---|---|---|---|---|---|---|---|---|---|---|---|
| | $MCA_u$ | $MCA_s$ | $H$ | $MCA_u$ | $MCA_s$ | $H$ | $MCA_u$ | $MCA_s$ | $H$ | $MCA_u$ | $MCA_s$ | $H$ |
| PREN[17] | 32.5 | 55.8 | 43.1 | 32.4 | 88.6 | 47.4 | - | - | - | 35.4 | 27.2 | 30.8 |
| LFGAA[10] | 43.4 | 79.6 | 56.2 | 50.0 | 90.3 | 64.4 | - | - | - | 20.8 | 34.9 | 26.1 |
| AREN[13] | 63.2 | 69.0 | 66.0 | 54.7 | 79.1 | 64.7 | 30.0 | 47.9 | 36.9 | 40.3 | 32.3 | 35.9 |
| RGEN[14] | 73.5 | 60.0 | 66.1 | 76.5 | 67.1 | **71.5** | 48.1 | 30.4 | 37.2 | 31.7 | 44.0 | 36.8 |
| DAZLE[8] | 56.7 | 59.6 | 58.1 | 60.3 | 75.7 | 67.1 | - | - | - | 52.3 | 24.3 | 33.2 |
| SELAR[16] | 43.0 | 76.3 | 55.0 | 32.9 | 78.7 | 46.4 | - | - | - | 23.8 | 37.2 | 29.0 |
| **DPPN** | 66.2 | 74.1 | **69.9** | 60.3 | 81.6 | 69.4 | 35.5 | 57.5 | **43.9** | 48.7 | 33.5 | **39.7** |

Table 4: Results of GZSL on four classification benchmarks using input size $224 \times 224$. The best result is bolded, and the second best is underlined.

## 5 Results of w/o Finetuning

We adopt a two-step training schedule that first trains DPPN with the fixed ResNet-101 backbone and then fine-tunes the whole network. As shown in Table 5, we also provide the results without finetuning. Our DPPN outperforms the best previous method by respectively $5.0\%$, $5.1\%$, $6.5\%$, and $0.9\%$ for H on CUB, AWA2, aPY, and SUN datasets. The results show that our DPPN is superior to previous methods even without finetuning.

| Methods | CUB | | | AWA2 | | | aPY | | | SUN | | |
|---|---|---|---|---|---|---|---|---|---|---|---|---|
| | $MCA_u$ | $MCA_s$ | $H$ | $MCA_u$ | $MCA_s$ | $H$ | $MCA_u$ | $MCA_s$ | $H$ | $MCA_u$ | $MCA_s$ | $H$ |
| MLSE[6] | 22.3 | 71.6 | 34.0 | 23.8 | 83.2 | 37.0 | 12.7 | 74.3 | 21.7 | 20.7 | 36.4 | 26.4 |
| CosineSoftmax[9] | 47.4 | 47.6 | 47.5 | 56.4 | 81.4 | 66.7 | 26.5 | 74.0 | 39.0 | 36.3 | 42.8 | 39.3 |
| DAZLE[8] | 56.7 | 59.6 | 58.1 | 60.3 | 75.7 | 67.1 | - | - | - | 52.3 | 24.3 | 33.2 |
| **DPPN** w/o ft. | 60.4 | 66.1 | **63.1** | 61.8 | 86.8 | **72.2** | 38.8 | 55.0 | **45.5** | 45.0 | 36.2 | **40.2** |

Table 5: Comparison results under without finetuning setting. The best result is bolded.

# 6 Different Combinations of Representations in Inference

Only the visual representation $f^K(X)$ at the $K$-th iteration is used in inference for simplicity. In this section, we conduct experiments of using different combinations of representations as the final image representation in inference, of which the results are shown in Table 6. From the table, we can see that using only the $K$-th representation obtains comparable results with summing up all the representations. Since the representation derives from the preceding representation, the preceding representations bring limited supplement to the final performance. Besides, concatenating all the representations results in a final representation with a large dim, which is harder to train and obtains worse performance. Moreover, using $f^1(X)$ as the final image representation can obtain considerable performance. However, our DPPN learns more transferable and distinctive representations through updating, which will achieve better results if using the final representation in inference.

| Methods | CUB | | | AWA2 | | | aPY | | | SUN | | |
|---|---|---|---|---|---|---|---|---|---|---|---|---|
| | $MCA_u$ | $MCA_s$ | $H$ | $MCA_u$ | $MCA_s$ | $H$ | $MCA_u$ | $MCA_s$ | $H$ | $MCA_u$ | $MCA_s$ | $H$ |
| Sum All | 72.0 | 75.2 | 73.6 | 61.4 | 89.7 | 72.9 | 39.5 | 60.5 | 47.8 | 42.4 | 38.6 | 40.5 |
| Concat All | 69.3 | 73.2 | 71.2 | 60.5 | 87.1 | 71.4 | 39.3 | 60.0 | 47.5 | 41.5 | 38.2 | 39.8 |
| $f^1(X)$ (1-th) | 65.8 | 80.3 | 72.3 | 62.9 | 82.8 | 71.5 | 37.9 | 58.9 | 46.1 | 46.3 | 34.4 | 39.5 |
| $f^K(X)$ ($K$-th) | 70.2 | 77.1 | 73.5 | 63.1 | 86.8 | 73.1 | 40.0 | 61.2 | 48.4 | 47.9 | 35.8 | 41.0 |

Table 6: Results of using different combinations of representations as the final image representation for inference.

# 7 Visualization of Representations

To intuitively demonstrate the distinction of our visual representations, we compare the representations of DPPN and Base-V2S using t-SNE on randomly selected categories, of which the results are shown in Fig. 1. Compared to Base-V2S, the representations are more compact inside a category and the margins between seen and unseen categories are enlarged in DPPN. This reflects that DPPN obtains more distinctive and transferable representations through progressively learning attribute-region correspondence and category discrimination.

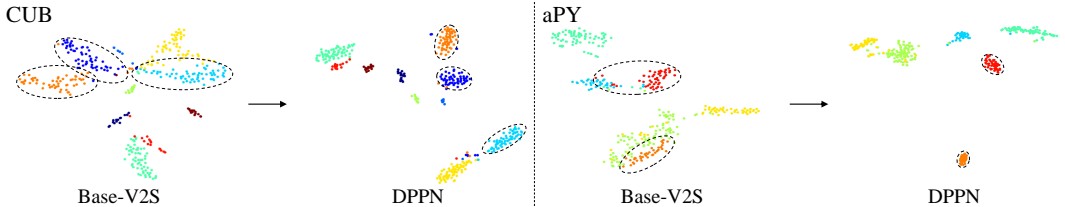

Figure 1: Visualization of representations on CUB and aPY datasets. Dotted circles denote unseen categories while seen categories are not circled.

# 8 Broader Impact

This paper proposes a novel method DPPN to tackle the domain shift problem in Generalized Zero-Shot Learning. The positive impacts of this paper can be summarized in two-fold: a) The proposed method enables people to recognize novel categories without elaborate labeling, which can reduce

annotation costs and difficulties. b) Our proposed DPPN brings negligible additional computation over the feature extractor network, which can save computing resources. The potential negative societal impacts contain: a) Our DPPN can alleviate the burden of labeling data, which may cause unemployment of data annotators. b) Our DPPN may be not suitable for data with large #categories and small #images in each category. c) The recognition of unseen domain remains still not good enough for industrial use, which should be further studied.