# OpenReview forum: "Dual Progressive Prototype Network for Generalized Zero-Shot Learning"
_NeurIPS.cc/2021/Conference — NeurIPS 2021 Poster_

### Official Review · Reviewer_LSav · 2021-07-09

**Rating:** 6
**Confidence:** 5

**Summary:**

This paper proposes a Dual Progressive Prototype Network that updates the prototypes for each attribute and categories progressively. To achieve this, an alternation updating strategy is adopted to adjust prototypes according to input images. The model is evaluated on 4 standard ZSL datasets and obtains the new state-of-the-art performance. The effect of each component is evaluated with extensive ablation study.


**Limitations And Societal Impact:**

However, I have the following concerns towards the network architecture, loss function, and the training procedure.

1. Line 215: is the architecture of the ResNet101 backbone changed? Why $C = 512$? Should it be $2048$?

2. Why not presenting the ZSL performance as well? The ZSL performance can present knowledge transferability as well.

3. Line 248: Any insights on why the localization ability will be more and more precise? Why the prototypes have self-correction ability? Apart from the qualitative results, a quantitative evaluation will be more convincing.

4. Equation 9: it seems that the loss is an ordinary cross-entropy loss. It might be inappropriate to call it contrastive loss.

**Main Review:**

The paper is well written and easy to follow. The alternative updated prototype can better capture the local information of each attribute and result in better ZSL knowledge transfer. The extensive experiments and ablation study validate the effectiveness of the proposed method.

[Post rebuttal] The authors have addressed my concern. After reading other reviewers' comments I still lean to accept the paper as the alternative updated prototype seems interesting to the community.

**Time Spent Reviewing:**

4

---

> ### Author Response · Authors · 2021-08-10
> **Authors' response**
>
> Thanks for your helpful comments. Below are detailed responses to your concerns.
>
> **Q1: Backbone feature dim.**
>
> **A1:** Since the feature dim ($C=2048$​​) from ResNet101 is too large for PAL, we use a conv. layer to reduce $C$​​ from $2048$​​ to $512$​​ for computation reduction. We are sorry for the negligence of this point in our manuscript and will add the details in the revision.
>
>
>
> **Q2: Adding Results of CZSL.**
>
> **A2:** Since DPPN is designed for the domain bias problem in GZSL, we only give the GZSL results in our manuscript (in CZSL, the image domain is prior-known). Here, we give the results under conventional ZSL setting below. $MCA_u$​​​ is used as evaluation metric.
>
> |         Method         |   CUB    |   AWA2   |   aPY    |   SUN    |
> | :--------------------: | :------: | :------: | :------: | :------: |
> |     AREN [Xie2019]     |   71.8   |   67.9   |   39.2   |   60.6   |
> | CosineSoftmax [Li2019] |   54.4   |   71.1   |   38.0   |   62.6   |
> |     RGEN [Xie2020]     |  *76.1*  | **73.6** |  *44.4*  | **63.8** |
> |   DAZLE [Huynh2020]    |   65.9   |    -     |    -     |    -     |
> |   GEM-ZSL [Liu2021]    | **77.8** |   67.3   |    -     |  *62.8*  |
> |      APN [Xu2020]      |   72.0   |   68.4   |    -     |   61.6   |
> |          DPPN          | **77.8** |  *73.3*  | **45.1** |   61.5   |
>
> As DPPN focuses on improving feature discrimination to alleviate the domain bias problem in GZSL, its performance on CZSL is not much impressive. However, DPPN still obtains comparable and even better performance compared to brand-new methods. We will add these results in the revision.
>
>
>
> **Q3: Insights on part localization ability and quantitative evaluation.**
>
> **A3:**  DPPN actually aggregates attribute-related clues from the visual features of a targeted image into attribute prototypes, so that the updated attribute prototypes can be corrected. This makes the prototypes adapted to the target image and produce better attribute localization.
>
> Following the evaluation settings in APN [Xu2020], we test our DPPN with the evaluation metric PCP on CUB. Mean PCP of DPPN is $86.4\%$, which is better than previous methods ($61.5\%$ of SGMA [Zhu2019] and $84.7\%$ of APN [Xu2020]).
>
>
>
> **Q4: About the contrastive loss.**
>
> **A4:** Thanks for your suggestion, we will revise this point.

---

### Official Review · Reviewer_LFjT · 2021-07-09

**Rating:** 6
**Confidence:** 4

**Summary:**

A Dual Progressive Prototype Network (DPPN) approach is proposed to progressively explore both attribute-region correspondence and category discrimination for GZSL. It alternatively locates attribute-related visual regions and updates attribute prototypes, which improves attribute localization ability and cross-domain transferability of visual representations.

**Limitations And Societal Impact:**

Yes

**Main Review:**

It's reasonable to alleviate domain shift problem by progressively adjusting the attribute prototypes and enlarging the category margins. The paper is logical and easy to understand. However, there are some details should be further stated and analyzed.
1.	Some of the descriptions in this paper is not accurate enough, e.g., for GZSL, the search space should be notified.
2.	In the ablation study, the accuracy of model with PCC and PAL on the seen samples is descended, and it seems that this method restrains the performance of classification on seen samples. I wonder the reason of the phenomenon, and the authors should explain it. Besides, the authors only discuss the ablation study on CUB and aPY, and I am curious about the results on AWA and SUN. It would be better if the authors shows it more completely.
3.	Besides the APN, whether the other competitive methods are bound up with DPPN? If they are, the authors should state mean idea of them and the similarities and differences between them and DPPN.
4.    The performance on the SUN dataset is not good actually, comparing with the recent work:
Zongyan Han, Zhenyong Fu, and Jian Yang. Learning the redundancy-free features for generalized zero-shot object recognition. In CVPR, pages 12862–12871, 2020.
Shaobo Min, Hantao Yao, Hongtao Xie, Chaoqun Wang, Zheng-Jun Zha, and Yongdong Zhang. Domain-aware visual bias eliminating for generalized zero-shot learning. In CVPR, pages 12661–12670, 2020
Wenjia Xu, Yongqin Xian, Jiuniu Wang, Bernt Schiele, and Zeynep Akata. Attribute prototype network for zero-shot learning. In NeurIPS, pages 21969–21980, 2020.



**Time Spent Reviewing:**

3

---

> ### Author Response · Authors · 2021-08-10
> **Authors' response**
>
> Thanks for your helpful comments. Below are detailed responses to your concerns.
>
> **Q1: Descanted accuracy in the seen domain.**
>
> **A1:** Thanks for this question. In ZSL, it has a phenomenon that improving feature transferability may harm the feature discrimination, because attribute labels of different categories are not orthometric to each other (different from one-hot vector of category labels). For example, the attribute vectors of leopard are very similar to those of tiger, which makes their features not discriminative. Thus, when a model is better aligned with category attributes, its transferability will be enhanced (unseen recognition improves) but the discrimination is harmed (seen recognition goes down). This is detailly illustrated in DVBE [Min2020]. Thus, in some benchmarks, our DPPN can preserve feature transferability, while improving feature discrimination, and in other benchmarks, DPPN can significantly improve transferability, while sacrificing fewer seen domain accuracy than other methods. For example, in CUB, the seen domain accuracy goes down, but unseen domain accuracy significantly improves. Both two situations are reasonable, because DPPN always pursues a better trade-off between transferability and discrimination of features to achieve better overall H.
>
>
>
> **Q2: Ablation study on AWA2 and SUN.**
>
> **A2:** We showed the results of ablation study on CUB and aPY for quick evaluation and space-saving in our manuscript. Here, the ablation study results on AWA2 and SUN are shown below.
>
> |  Method  | $\mathcal{L}_{v2s}$ | $\mathcal{L}_{cl}$ | $\mathcal{L}_{sa}$ | $K$  | AWA2 $MCA_u$​ | AWA2 $MCA_s$​ | AWA2 $H$​ | SUN $MCA_u$​ | SUN $MCA_s$​ | SUN $H$​ |
> | :------: | :-----------------: | :----------------: | :----------------: | :--: | :----------: | :----------: | :------: | :---------: | :---------: | :-----: |
> | Base-V2S |          √          |                    |                    |  -   |     50.6     |     78.7     |   61.6   |    36.8     |    24.0     |  29.0   |
> |   +PCC   |                     |         √          |                    |  1   |     51.1     |     80.3     |   62.4   |    39.2     |    28.4     |  32.9   |
> |   +PAL   |                     |                    |         √          |  1   |     54.1     |     81.3     |   64.9   |    41.3     |    31.1     |  35.5   |
> | +PCC&PAL |                     |         √          |         √          |  1   |     54.5     |     84.1     |   66.1   |    45.8     |    34.4     |  39.3   |
> | +PCC&PAL |                     |         √          |         √          |  3   |     63.1     |     86.8     |   73.1   |    47.9     |    35.8     |  41.0   |
>
>
> | $f_{ar}(\cdot)$ | $cat(\cdot)$ | $sum(\cdot)$ | $f_{cs}(\cdot)$ | AWA2 $MCA_u$​ | AWA2 $MCA_s$​ | AWA2 $H$​ | SUN $MCA_u$​ | SUN $MCA_s$​ | SUN $H$​ |
> | :-------------: | :----------: | :----------: | :-------------: | :----------: | :----------: | :------: | :---------: | :---------: | :-----: |
> |        √        |      √       |              |        √        |     63.1     |     86.8     |   73.1   |    47.9     |    35.8     |  41.0   |
> |                 |      √       |              |        √        |     62.3     |     85.2     |   72.0   |    47.5     |    34.6     |  40.0   |
> |        √        |              |      √       |        √        |     62.0     |     79.9     |   69.8   |    44.5     |    36.0     |  39.8   |
> |        √        |      √       |              |                 |     61.2     |     85.3     |   71.3   |    45.3     |    36.7     |  40.5   |
>
>
> | $\lambda$ | AWA2 $MCA_u$​ | AWA2 $MCA_s$​ | AWA2 $H$​ | SUN $MCA_u$​ | SUN $MCA_s$​ | SUN $H$​  |
> | :-------: | :----------: | :----------: | :------: | :---------: | :---------: | :------: |
> |     0     |     58.4     |     82.2     |   68.3   |    44.4     |    32.9     |   37.8   |
> |    0.5    |     62.7     |     84.3     |   71.9   |    47.6     |    35.4     |   40.6   |
> |     1     |     63.1     |     86.8     | **73.1** |    47.9     |    35.8     |   41.0   |
> |     2     |     62.9     |     85.6     |   72.5   |    48.7     |    41.2     | **41.2** |
> |     5     |     62.4     |     85.4     |   72.1   |    46.5     |    35.4     |   40.2   |
> |    10     |     61.9     |     84.9     |   71.6   |    46.8     |    34.5     |   39.7   |
>
>
> |  $N_v$​​   | AWA2 $H$​ |  $N_v$​   | SUN $H$​  |
> | :------: | :------: | :------: | :------: |
> |   595    |   69.8   |   612    |   39.9   |
> |   1105   |   72.4   |   1122   |   40.3   |
> | **2125** | **73.1** | **2142** | **41.0** |
> |   4165   |   68.5   |   4182   |   40.1   |
>
> In the ablation study of $\lambda$​​, the best $\lambda=1.0$​ for CUB, AWA2, and aPY datasets and $\lambda = 2.0$​ for SUN. Thus, overall, we set $\lambda=1.0$​​​ for the four datasets.
>
> We will add these results in the revision, as well as the visualization.
>
>
>
> **Q3: Other competitive methods that are bound up with DPPN.**
>
> **A3:** Similar to DPPN, DAZLE [Huynh2020] and AREN [Xie2019] also target to localize attribute-related regions to produce image features. The main difference lies in that DAZLE and AREN utilize dense attention mechanism to directly infer attention maps, while DPPN learns attribute prototypes that are dynamically updated for different images to localize attribute regions, which is more interpretable and obtains better results in Table 3 of the manuscript. The main idea of DPPN, APN, DAZLE, and AREN is to localize attribute regions to produce image features. We will analyze more methods in Sec. 3.5 Discussion.
>
>
>
> **Q4: Performance on SUN is not good.**
>
> **A4:** Indeed, the performance of DPPN on SUN is not as impressive as on the other three benchmarks. The reason has been given on Line 307-309 on page 9 in our manuscript. Notably, compared with the most related methods, e.g., APN, DAZLE, AREN, that explore attribute localization ability, DPPN shows great superiority and good potential of this idea. The high results on SUN in APN [Xu2020] are actually based on a generative method which utilizes extra synthesized training data. For fair comparisons, our DPPN actually surpasses the embedding-based APN model by $3.4\%$​​​​​​​ on SUN.
>
>
>
> **Q5: Some descriptions are not accurate enough.**
>
> **A5:** Thanks for your suggestion. We will carefully polish our descriptions in the revision.

---

> > ### Comment · Reviewer_LFjT · 2021-08-21
> > **Re: Rebuttal response**
> >
> > I appreciate the great efforts that the authors have made in response to my questions and concerns. The revision clarifies almost all the points I raised. I recommend to the acceptance of this paper.

---

### Official Review · Reviewer_UvLd · 2021-07-12

**Rating:** 6
**Confidence:** 3

**Summary:**

This paper aims for generalised zero-shot learning with prototypes. Rather than using one set of attributes, two sets are used here: attribute localization prototypes and category-level prototypes. Experiments on four datasets show the effect of the proposed approach, with top scores for harmonic means.

**Limitations And Societal Impact:**

The checklist mentions Section 4.3 for a discussion on limitations, but it only covers praise for the method, except for a footnote on SUN. The societal impact statement notes that fewer annotations due to zero-shot generalization will lead to unemployed annotators.

**Main Review:**

Strengths:

The first strength of the paper is the notion of two levels of prototypes. Prototypes have recently shown to be effective for (generalised) zero-shot learning but mostly focus on attribute learning. Here, prototypes are learned both at the attribute-level and category-level, to both help with finding suitable attribute representations and class representations.

Second, the paper is accompanied by many ablation studies to better understand the method and its parameters. Moreover, the comparative evaluation shows the effectiveness of the proposed approach, with state-of-the-art harmonic means on all four datasets.

Third, the figures in the paper help with understanding the method and the difference with related research. The discussion on the relation with priors works is also extensive, which helps with placing the paper into the proper context.

Weaknesses:

The first weakness of the paper is that the method is not well explained. While the method is not overly complicated, the writing and explanation do not help with understanding what is going on. At a high-level, this paper follows APN with category-level prototypes on top. In multiple cases, the paper makes things harder to understand however. These include:

- The paper states that “APN [47] constructs prototypes for separate attributes, which are shared by all images to localize attribute-related regions via region searching. In this paper, instead of using prototypes shared by all images, DPPN dynamically adjusts attribute prototypes according to different images” (lines 91-93) and “First, instead of using shared prototypes, DPPN progressively adjusts attribute prototypes according to different images via the novel PAL.” (Line 194-195). What does this even mean? According to Equation 7, the proposed loss is over all images, just like in APN. It is not clear what is precisely novel in the first part of the method based on the paper.
- How is the progressive learning done? Is there a stage-wise training of the attribute prototypes first, then the same for the category prototypes, and finally the overall loss is optimised? Or is everything optimised end-to-end? Any explanation on how the actual training is done is missing in the paper.
- “Similar to Lsa, the multi-stage supervision is used in Lcl.” What does multi-stage supervision mean?

A conceptual difference with APN is the use of category attributes. The loss in Equation 9 pushes representations from different categories further away. While this makes sense in a supervised setting, it poses a problem in the zero-shot setting, as it becomes harder to have an unseen-class be the most likely category. This is also evident from the experiments. While the harmonic mean is high overall, this is clearly due to a better recognition of the seen categories. The accuracy of the unseen categories is not great across the datasets compared to other approaches. The harmonic mean is saved by the improved recognition for seen categories. As such, it seems that the approach is more tailored towards supervised learning than zero-shot learning. Table 1 makes things harder to understand, since the results between CUB and aPY are anything but consistent.For CUB the seen accuracy actually goes down, while it forms the biggest improvement on aPY.

**Time Spent Reviewing:**

4

---

> ### Author Response · Authors · 2021-08-10
> **Authors' response**
>
> Thanks for your helpful comments. Below are detailed responses to your concerns.
>
>
>
> **Q1: Confusing explanation about progressive attribute prototypes and comparison with APN.**
>
> **A1:** The **common ground** between DPPN and APN is that both of them learn a set of attribute prototypes for a dataset. Taking CUB dataset ($312$ attributes) as an example, the attribute prototypes are trainable parameters of dim=$C\times 312$ ($C$ is backbone feature channel), which are trained via back-propagation and shared by all images in CUB. The **different point** is that, when testing an image, APN directly uses the well-trained attribute prototypes to localize attribute-related regions for an image. Differently, DPPN first adjusts attribute prototypes according to visual contents of the testing image, and then uses these adjusted attribute prototypes to localize regions. Usually, the attribute prototypes will be adjusted several times (we found that adjusting three times obtains the best performance), thus we call DPPN “progressively adjusts attribute prototypes for an input image”. Based on such a progressive adjusting process, the final attribute prototypes for different images are different in DPPN, which we call “dynamically adjust attribute prototypes for different images”. We are much sorry for this confusion and will refine our presentation in the revision.
>
>
>
> **Q2: About training process: a) Multi-stage loss and b) two-step training.**
>
> **A2:** a) The PAL and PCC are trained together. As illustrated in A1, DPPN progressively adjusts attribute (and category) prototypes several times, and we regard each updating process as a stage. At the $k$​​​-th iteration stage, there are two losses, e.g., $L_{sa}^{k}$​​​ for attribute prototypes and $L_{cl}^{k}$​​​ for category prototypes. During training, the final loss of DPPN is to sum up losses of all stages $L_{all}\leftarrow\sum_{k=1}^{K} (L_{sa}^{k}+L_{cl}^{k})$​​​. All parameters in DPPN are end-to-end trained by $L_{all}$​​​. {$L_{sa}^{1}+L_{cl}^{1}$​, $L_{sa}^{2}+L_{cl}^{2}$​, ... , $L_{sa}^{K}+L_{cl}^{K}$​} is called multi-stage (K stages) supervision.
>
> b) Actually, instead of two-stage training, it is more accurate to call our training process as two-step training. In the first step, we warm up DPPN by only training the head with fixed backbone for several epochs. At the second step, we fine-tune the whole network of DPPN. Loss $L_{all}$​​​​ is used for training both two steps. We will explain these points more clearly in our revision.
>
>
>
> **Q3: Unseen-class accuracy is not great.**
>
> **A3:** Thanks for this question. In GZSL, the performance on MCA_s and MCA_u is not always consistent. For example, RGEN [Xie2020] obtains ($MCA_u=73.5$, $MCA_s= 60.0$) on CUB and  ($MCA_u=31.7$, $MCA_s=44.0$) on SUN, which has inconsistent improvements on different datasets. One reason is that the post-processing operation (calibration) will balance $MCA_u$ and $MCA_s$ to obtain an overall high H. And another reason is that improving feature transferability may harm the feature discrimination, because attribute labels of different categories are not orthometric to each other (different from one-hot vector of category labels). When a model is better aligned with category attributes, its transferability will be enhanced but the discrimination is harmed. This is detailly illustrated in DVBE [Min2020]. Thus, in some benchmarks, our DPPN chooses to preserve feature transferability, while improving feature discrimination for higher $H$. And in other benchmarks, DPPN significantly improves transferability, while sacrificing fewer seen domain accuracy than other methods. For example, on CUB, the seen domain accuracy goes down, but unseen domain accuracy significantly improves. Both two situations are reasonable, because DPPN always pursues a better trade-off between transferability and discrimination of features to achieve better overall $H$.

---

> > ### Comment · Reviewer_UvLd · 2021-09-01
> > **End of rebuttal**
> >
> > The rebuttal has helped me to better understand the difference with APN. I strongly suggest to be more precise in explaining your own method. Sentences like this remain handwaivy: "DPPN first adjusts attribute prototypes according to visual contents of the testing image, and then uses these adjusted attribute prototypes to localize regions."
> >
> > Since the ratings diverged, there has been some discussion amongst reviewers. After the discussion, I remain at a score above the threshold.

---

### Official Review · Reviewer_ZcLN · 2021-07-19

**Rating:** 6
**Confidence:** 4

**Summary:**

The bias towards seen categories in GZSL is a main challenge, and the authors argue that shared prototypes across all images are partly responsible, as well as providing suboptimal discriminability. Thus, the paper proposes increasing the discriminability of attributes and categories prototypes in a per-image basis via progressive refinement through an iterative attention mechanism. The results show improvement over related works, such as AREN and APN.

**Limitations And Societal Impact:**

The authors didn't discussed these topics in the manuscript.
Regarding limitations of their work and suggestions, please refer to my comments above. About societal impact, I don't foresee any significant negative one.

**Main Review:**

Originality:
The paper borrows from previous frameworks and the main novelty is the progressive adaptation of prototypes to each image, which is interesing. Overall, the novelty is moderate.

Quality:
The submission is technically sound, with exhaustive experiments.
Yet widely used, I'm concerned about the harmonic mean (H) metric not being an accurate representation of the performance of a GZSL method, since it requires calibration, and thus can be unfair to compare non-calibrated approaches, as discussed in [Changpinyo2020]. The seen-unseen accuracies curve and AUSUC metric described in [Changpinyo2020] are arguably better metrics.

Clarity:
The submission is relatively clear and well organized. However, I have some concerns:
- Not clear if the method uses a part extractor or just operates on the convolutional feature tensor. Regions refers to extracted bounding boxes, or implicit local regions corresponding to receptive fields of representations?
- Not clear how local features are aggregated into the global representation. It usually uses GAP, but here is removed and not clarified afterward (or couldn't find it).
- Please don't use comma to separate thousands when enumerating numebers, e.g. N_v=2, 496, 2, 125, 2, 048, 2, 142. It is very confusing.

Significance:
The authors propose essentially an improvement over a previous GZSL framework. The results provide a moderate improvement over the baseline (i.e. shared prototypes), but I think the significance may be low since I think the problem of discrimination lies less in the alignment of prototypes, and more in the aggregation mechanism.

I have several concerns about the paper:
- The paper focuses on improving the discriminability of attribute prototypes, but not on how they are aggregated in the global representation (which is not clear to me from the text). Recently, SELAR showed that an important bottleneck in GZSL to benefit from localized attribute representations is precisely the aggregation mechanism via GAP, showing a large improvement in discriminability for GZSL simply replacing GAP in ResNet by global max pooling. Thus, the gain of the propose approach may be only significant with a suboptimal aggregation mechanism (e.g. GAP or other).
- Thus, the authors should compare with SELAR, which is a straightforward to implement baseline, and then implement their approach to evaluate whether progressive prototype improvement is still significant when a more discriminative aggregation mechanism such as GMP is used.
- The comparison may not be fair to many methods. The proposed method requires a large image of 448 x 448 (as AREN), but most papers work on 224x224 pixels (e.g. SELAR). It also requires fine tuning (perhaps also a part extractor??), while other methods use non fine tuned feature extractors. For a clear and fair comparison, all these details should be explicitly mentioned.

[Chang2020] Classifier and Exemplar Synthesis for Zero-Shot Learning, IJCV 2020
[SELAR] On Implicit Attribute Localization for Generalized Zero-Shot Learning, SPL 2021

**Time Spent Reviewing:**

5

---

> ### Author Response · Authors · 2021-08-10
> **Authors' response**
>
> Thanks for your helpful comments and agreement with our novelty of progressive prototypes. Besides progressive prototypes, we also introduce a novel aggregation mechanism for attribute-region correspondence preservation. Below are detailed responses to your concerns.
>
> **Q1: About contributions and aggregation mechanism.**
>
> **A1:** Actually, PAL module in our DPPN has two core technical contributions: a) **a new progressive prototype mechanism** for attributes; and b) **a new aggregation mechanism:** DPPN uses the progressively updated attribute prototypes (e.g., $312$ attribute prototypes on CUB) to localize local features in the backbone feature map (size $512\times17\times17$) and produces a $512\times312$ feature matrix. As introduced on Line 147-150 in our paper, the feature matrix contains specific visual patterns for different attributes. Thus, DPPN further maps the $512\times312$​​​ feature matrix into $8\times312$​​​ via FC, and concatenates the $312$​​​ local features into a global one of dim=$2496$​​​ ($8*312$​​​​) to represent an image. Such an aggregation mechanism is totally different from the global average pooling in APN [Xu2020] or global max pooling in SELAR [Yang2021], and can better preserve attribute-region correspondence. The third row of Table 2 in our ablation study also proves the superiority of this aggregation mechanism over the widely-used summing up operation. We are sorry that the aggregation mechanism is not emphasized in the introduction, since it is a part of our PAL module. As we totally agree that the aggregation mechanism is much significant in improving the feature discrimination, we will emphasize more on this point in our revised version.
>
>
>
> **Q2:  Comparison with SELAR and implement their aggregation to evaluate the improvement of progressive prototypes.**
>
> **A2**: As in A1, our DPPN also provides a novel aggregation mechanism. Since our DPPN converts the backbone feature map (e.g., $512\times17\times17$) to the attribute-related matrix (e.g., $512\times 312$ on CUB), we directly use a max pooling along the $312$ local features to implement SELAR in DPPN. The results are shown below. (The values are given in the form of $H$ ($MCA_u$, $MCA_s$))
>
> | Aggregation Mechanism |        CUB​        |        aPY​        |
> | :-------------------: | :---------------: | :---------------: |
> |     $sum(\cdot)$      | 68.7 (67.0, 70.5) | 45.2 (35.6, 62.0) |
> |     $max(\cdot)$      | 71.7 (70.4, 73.0) | 45.8 (37.3, 59.4) |
> |  DPPN ($cat(\cdot)$)​  | 73.5 (70.2, 77.1) | 48.4 (40.0, 61.2) |
>
> The results show that with max aggregation, our DPPN performs better than widely-used summing up. This proves that $max(\cdot)$ can aggregate local information better than $sum(\cdot)$. However, using concatenation to aggregate local region features obtains the best performance among the three aggregation mechanisms, which is because that the aggregation mechanism proposed in our DPPN explicitly preserve the attribute-region correspondence. We will add the results in our revision.
>
>
>
> **Q3: The method uses a part extractor or just operates on the convolutional feature tensor?**
>
> **A3**: DPPN directly operates on the convolution feature tensor (e.g., a $512\times17\times17$​​ feature map extracted by the ResNet101 backbone + a conv. layer). The visual regions implicitly refer to the local regions corresponding to receptive fields of representations.
>
>
>
> **Q4: Non-calibration $H$​​ and AUSUC results.**
>
> **A4**: We actually have given the non-calibration results between DPPN and related methods in Table 1 of the supplementary pdf. As shown in that results, our DPPN outperforms the best one by respectively $15.3\%$​, $8.8\%$​, and $7.3\%$​ for $H$​ on CUB, AWA2, aPY datasets, and obtains comparable performance on SUN dataset. Please refer to Line 5-10 of the supplementary pdf for the detailed results and analysis.
>
> Besides, following your suggestion, we also conduct experiments with recent related methods reporting results on AUSUC metric, of which the results are shown below:
>
> |        Method         | CUB AUSUC | AWA2 AUSUC | aPY AUSUC | SUN AUSUC |
> | :-------------------: | :-------: | :--------: | :-------: | :-------: |
> | SYNC [Changpinyo2016] |   33.7    |    50.4    |     -     |   24.1    |
> |  COSMO [Atzmon2019]   |   35.7    |     -      |     -     |   23.9    |
> | EXEM [Changpinyo2020] |   36.6    |    55.9    |     -     | **25.1**  |
> |         DPPN          | **56.3**  |  **63.6**  | **33.4**  |   23.1    |
>
> As shown in the results, our DPPN outperforms the best one by respectively $19.7\%$ and $7.7\%$ for AUSUC on CUB and AWA2 datasets, and obtains comparable performance on SUN dataset. The robust improvements over various metrics prove that DPPN can effectively alleviate the domain bias problem in GZSL.
>
>
>
> **Q5: Results of input size $224\times224$.**
>
> **A5**: For fair comparisons with recent SOTA methods, we use the setting of input size $448\times 448$​ following recent methods, e.g., VSE-S [Zhu2019], GEM-ZSL [Liu2021], and AREN [Xie2019]. Thanks for your suggestion, we further conduct experiments comparing DPPN with recent methods that use $224\times 224$​ as input size. The results are shown below. (The values are given in the form of $H$​ ($MCA_u$​, $MCA_s$​))
>
> |        Method         |          CUB          |         AWA2          |          aPY​          |          SUN          |
> | :-------------------: | :-------------------: | :-------------------: | :-------------------: | :-------------------: |
> |     PREN [Ye2019]     |   43.1 (32.5, 55.8)   |   47.4 (32.4, 88.6)   |           -           |   30.8 (35.4, 27.2)   |
> |    LFGAA [Liu2019]    |   56.2 (43.4, 79.6)   |   64.4 (50.0, 90.3)   |           -           |   26.1 (20.8, 34.9)   |
> |    AREN [Xie2019]     |   66.0 (63.2, 69.0)   |   64.7 (54.7, 79.1)   |   36.9 (30.0, 47.9)   |   35.9 (40.3, 32.3)   |
> |    RGEN [Xie2020]     |  *66.1* (73.5, 60.0)  | **71.5** (76.5, 67.1) |  *37.2* (48.1, 30.4)  |  *36.8* (31.7, 44.0)  |
> |   DAZLE [Huynh2020]   |   58.1 (56.7, 59.6)   |   67.1 (60.3, 75.7)   |           -           |   33.2 (52.3, 24.3)   |
> |   SELAR [Yang2021]    |   55.0 (43.0, 76.3)   |   46.4 (32.9, 78.7)   |           -           |   29.0 (23.8, 37.2)   |
> | DPPN ($224\times224$) | **69.9** (66.2, 74.1) |  *69.4* (60.3, 81.6)  | **43.9** (35.5, 57.5) | **39.7** (48.7, 33.5) |
>
> As shown in the Table, our DPPN outperforms the best previous method by respectively $3.8\%$​​​​​​​​​, $6.7\%$​​​​​​​​​, and $2.9\%$​​​​​​​​​ for $H$​​​​​​​​​​ on CUB, aPY, and SUN datasets. and achieves the second best performance on AWA2. The SOTA performance on different resolutions demonstrates the effectiveness and generalization of our DPPN. We will add the results in our revision.
>
>
>
> **Q6: W/o finetuning results.**
>
> **A6**: We adopt a two-step training schedule that first trains DPPN with the fixed ResNet backbone and then fine-tunes the whole network. Following your suggestion, we provide the results without finetuning as below. (The values are given in the form of $H$ ($MCA_u$, $MCA_s$))
>
> |         Method         |          CUB          |         AWA2​          |          aPY          |          SUN          |
> | :--------------------: | :-------------------: | :-------------------: | :-------------------: | :-------------------: |
> |    MLSE [Ding2019]     |   34.0 (22.3, 71.6)   |   37.0 (23.8, 83.2)   |   21.7 (12.7, 74.3)   |   26.4 (20.7, 36.4)   |
> | CosineSoftmax [Li2019] |   47.5 (47.4, 47.6)   |   66.7 (56.4, 81.4)   |  *39.0* (26.5, 74.0)  |  *39.3* (36.3, 42.8)  |
> |   DAZLE [Huynh2020]    |  *58.1*(56.7, 59.6)   |  *67.1* (60.3, 75.7)  |           -           |   33.2 (52.3, 24.3)   |
> |      DPPN w/o ft.      | **63.1** (60.4, 66.1) | **72.2** (61.8, 86.8) | **45.5** (38.8, 55.0) | **40.2** (45.0, 36.2) |
>
> As shown in the Table, our DPPN outperforms the best previous method by respectively $5.0\%$​​​​, $5.1\%$​​​​, $6.5\%$​​​​, and $0.9\%$​​​​ for $H$​​​​ on CUB, AWA2, aPY, and SUN datasets. The results show that our DPPN is superior to previous methods even without finetuning. We will add the results in our revision.
>
>
>
> **Q7: Limitation and societal impact.**
>
> **A7:** We have given the limitation on Line 307-309 in Sec. 4.3 which points out that DPPN may have difficulties to learn accurate attribute localization and category discrimination when \#images in each category is small. The societal impact is concluded on Line 23-27 in the Supplementary pdf.
>
>
>
> **Q8: Typos of commas.**
>
> **A8:** Thanks for your suggestion. We will revise the confusing commas in our final revision.

---

> > ### Comment · Reviewer_ZcLN · 2021-09-02
> > **Response to rebuttal**
> >
> > I appreciate the response and additional results from the authors. They addressed most of my concerns in terms of metrics and a more fair and clear comparison to evaluate the contribution of the proposed method. Measured by AUSUC the method still achieves the best result in three datasets, but underperforms in SUN. Measured by HM, aggregating with max pooling improves the discriminability over average pooling, as expected, but the proposed approach still has a noticeable gain. Measured by HM, the method is also very competitive on 224x224 images, surprisingly outperforming 448x448 significantly on CUB. Unfortunately, the authors didn't provide AUSUC results in the subsequent results, so my former concern about the significance of results measured using HM remains. I suggest the authors to include that metric in all the experiments.

---

> > > ### Author Response · Authors · 2021-09-07
> > > **Further results**
> > >
> > > Thanks for your appreciation and further suggestions. We will add the results measured by AUSUC for all experiments in our revision. Here, we provide the important results that you are concerned in terms of AUSUC.
> > >
> > >
> > >
> > > **Effect of different aggregation mechanisms.**
> > >
> > > | Aggregation Mechanism | CUB AUSUC | AWA2 AUSUC | aPY AUSUC | SUN AUSUC |
> > > | :-------------------: | :-------: | :--------: | :-------: | :-------: |
> > > |     $sum(\cdot)$      |   47.6    |    48.9    |   28.7    |   19.9    |
> > > |     $max(\cdot)$      |   50.9    |    52.0    |   30.1    |   21.5    |
> > > |  DPPN ($cat(\cdot)$)  | **56.3**  |  **63.6**  | **33.4**  | **23.1**  |
> > >
> > > The performance of AUSUC is consistent with $H$​. Our DPPN with $cat(\cdot)$​ aggregation still achieves the best performance in terms of AUSUC among these three kinds of aggregation mechanisms. The effectiveness of our $cat(\cdot)$​ aggregation is because that this aggregation explicitly preserves the attribute-region correspondence.
> > >
> > >
> > >
> > > **Input size $224\times224$ and w/o finetuning results​​.**
> > >
> > > We follow the methods SYNC [Changpinyo2016], COSMO [Atzmon2019], and EXEM [Changpinyo2020] to use the setting of input size $224\times224$​ and w/o ft. Results are shown below.
> > >
> > > |            Method             | CUB AUSUC | AWA2 AUSUC | aPY AUSUC | SUN AUSUC |
> > > | :---------------------------: | :-------: | :--------: | :-------: | :-------: |
> > > |     SYNC [Changpinyo2016]     |   33.7    |    50.4    |     -     |   24.1    |
> > > |      COSMO [Atzmon2019]       |   35.7    |     -      |     -     |   23.9    |
> > > |     EXEM [Changpinyo2020]     |   36.6    |    55.9    |     -     | **25.1**  |
> > > | DPPN ($224\times224$, w/o ft) | **45.1**  |  **57.3**  | **21.6**  |   22.8    |
> > >
> > > Our DPPN outperforms the best one by respectively $8.5\%$​ and $1.4\%$​​ on CUB and AWA2 datasets and achieves comparable results on SUN in terms of AUSUC. The superior results demonstrate that our DPPN is generalized and effective in smaller input size and without finetuning settings. The reason that DPPN does not achieve better results on SUN maybe that #categories is large while #images in each category is small in SUN, leading to difficulties for DPPN to learn accurate attribute localization and category discrimination.
> > >
> > >
> > >
> > > The experimental results of AUSUC prove that our DPPN can effectively alleviate the domain shift problem and achieve SOTA performance in GZSL through progressively exploring attribute-region correspondence and category discrimination.

---

### Decision · Program_Chairs · 2021-09-28

**Decision:**

Accept (Poster)

**Comment:**

This paper addresses generalised ZSL via proposing a new prototype refinement-based scheme to reduce confusion between known and novel categories. Reviewers felt there were various sources of confusion and explanation quality. However they also agree that the methodology is solid enough and the evaluation is strong. I recommend accept, and encourage the authors to take reviewers comments on board in refining the quality and clarity of exposition in the final version.

**Consistency Experiment:**

NeurIPS has a long history of experimentation. In 2014, NeurIPS ran an experiment in which 10% of submissions were reviewed by two independent committees to quantify the randomness in the review process. This year, we repeated a variant of this experiment to see how the quality of the review process has changed over time.  This paper was part of the experiment and was therefore assigned to two committees (consisting of reviewers, an Area Chair, and a Senior Area Chair) that reached independent decisions.  If both committees made the same recommendation, this recommendation was followed. If a single committee recommended acceptance, the paper was accepted (with the exception of a few cases in which the other committee identified what we considered a fatal flaw, e.g., an error in a key result).

This copy’s committee reached the following decision: **Accept (Poster)**

The other committee assigned to the paper recommended **Reject**.  You can find the other set of reviews, along with any follow up discussion with the authors here:
https://openreview.net/forum?id=BW2Z6B7S9KZ